# Research on Tracking the Behavior of the Ciobanus Forest Road over a Season Time through Specific Tests and Analysis

**Ioan Bitir [1], Rudolf Derczeni [1], Aurel Lunguleasa [2,*], Cosmin Spirchez [2] and Valentina Ciobanu [1]**

[1] Faculty of Forestry and Logging, Transilvania University of Brasov, B-dul Eroilor 29, 500360 Brasov, Romania; ioan.bitir@unitbv.ro (I.B.); derczeni@unitbv.ro (R.D.); ciobanudv@unitbv.ro (V.C.)

[2] Faculty of Furniture Design and Wood Engineering, Transilvania University of Brasov, B-dul Eroilor 29, 500360 Brasov, Romania; cosmin.spirchez@unitbv.ro

[*] Correspondence: lunga@unitbv.ro

**Abstract:** Forest roads are of great economic importance as they ensure the transport of logs and forest biomass toward collection and processing centers, which is why they should be evaluated periodically, in order to establish the degree of degradation and periodicity and rehabilitation methodology and procedures. The main purpose of the paper is to follow the behavior of the Ciobanus forest road through specific tests over a difficult season of 5 months, in order to diagnose the degree of surface wear and structure degradation. Regarding the traffic on this forest road, an exhaustive study was made during the 2013–2017 period, and for in situ or in laboratory tests a more complex study during the year 2018, in the March-June period was also made. Out of the total of 20 tests that evaluated the Ciobanus forest road, 5 of them were classified as appropriate and 15 unsuitable for traffic, meaning the forest road had to be completely rehabilitated. Moreover, it has been shown that this forest road is part of the category of secondary forest roads and needs a total overhaul to cope with the increasing traffic or tonnage of trucks. Through the methodology and the obtained results, the paper supports the specialists in the field of forest roads to be able to diagnose or evaluate such a road, and to realize a program and its timing for maintenance.

**Keywords:** forest road; dynamic penetration; granularity; consistency index; wear resistance

## 1. Introduction

Forest roads or forest tracks are the roads on which motor vehicles run and are used exclusively for forestry purposes, such as conservation or logging activities. Forest roads could sometimes be used for hiking or mountain biking, depending on the local rules of the road. When the forest on which the forest road is located is private and/or part of the conservation heritage, the road is provided with barriers and an inscription prohibiting access. Mountain forest roads are the most complex ones, compared to those in the plains or hills. First of all, they are wider than the ones in the plain/hill in order to circulate the trucks with trailer safely, and on the other hand, because they are more subject to erosion and landslides.

The materials used to make forest roads can be characterized from a physical or mechanical point of view. From a physical point of view, the materials used have the following characteristics: the porosity defined as the ratio between the pore volume and the total volume of the material (high for sands and gravels and low for clay soils), specific density (1650 kg/m$^3$ for sand, 1680 kg/m$^3$ for dust, and 1700 kg/m$^3$ for crushed stone and dry gravel), the moisture content defined as the ratio between the amount of water and the amount of dry material, with a degree of saturation of maximum 0.4 for dry materials and more than 0.8 for very wet materials, the plasticity defined as the degree of return after the application of a deformation force (having a plasticity index of 0 for sand and 20–35 for clay), the consistency index defined as the mode of behavior of different materials under various stress conditions (for the consistency index less than 0.75 the soil is sticked to the

bucket of the machine), the permeability of the soils that characterizes the pass of water through the pores of the material (the materials with fine particles have a higher porosity than those with larger particles), the capillarity or compaction capacity respectively the height at which the water rises in a capillary tube (a material with large gaps has a small capillarity) and the frequency of frost-thaw states, which leads to the destruction of rocks in winter period [1]. The mechanical characteristics of the materials used for forest roads are two. The first of them is compressibility, defined as the ability to compact the land and to be deformed under the action of external compression stresses, respectively to reduce the distance between particles, which depends very much on the particle size composition. The second mechanical property is the resistance to shear which can lead to cracks or fissures in the surface roadway of forest roads.

Some authors [1] reviewed the main studies of the last 20 years on forest road networks, with a focus on the impact of forest roads on hydrological processes. The hydrological basins in the area of forest roads were studied, especially during the period of abundant precipitations, the deposition of alluvium on the road and/or on the area of the main torrents. Other authors [2] analyzed the benefits of forest roads in wildfire management in the western USA. Of interest in this paper was how the forest road network supported the control, prevention, and management of forest fires. Đuka et al. [3] made an analysis of forest accessibility in order to harvest oak trees that fell by storms or dried on their feet (by defoliation). The paper took into account the spatial distribution of windfalls in the field and the volume of felled trees over a long period of seven years.

De Witt et al. [4] studied the degradation of forest roads in the Pacific Northwest, USA area, focusing on the process of degradation of aggregates used to build such a forest road. In this sense, by testing the wear rate of standard materials used in the realization of such forest roads, it was possible to make a correct prediction on the impact of a certain type of material on the road surface over time. Aricak in 2015 [5] analyzed the surface of forest roads affected by erosion, respectively conducted a case study in the Kastamonu region of Turkey. The studied dependence factors were the gradient of the road slope and the density of trees in the analyzed area using GeoEye satellite images and a digital elevation model. Cavalli and Grigolato 2010 [6] studied the influence of the characteristics of forest road networks on the costs of chopping and chipping obtained in forestry operations in forest road areas. Costs have also been studied when extending/rehabilitation an existing forest road. Gumus et al. [7] considered some aspects of the impact of the realization of a forest road for log harvesting on the environment. In order to set up additional networks of forest roads, a geographical information system was used to evaluate the data obtained from the design of the new 16 secondary roads with minimal impact on the environment. Parsakhoo et al. [8] measured road erodibility and showed that the surface material of the forest road had an erosion of 2.3 times higher than the fill slope of the road and of 1.3 times higher than the cut slope. Texture, bulk density, moisture content and organic matter from topsoil, cut slope, fill slope, and forest ground were determined for comparison. Pazhouhan et al. [9] used the ground-penetrating radar method to detect the subsoil rock existing in the infrastructure of the forest road. A comparison between the existing forest road and the forecast for a new road was made successfully. Akay et al. [10] started their study from the premise that in order to ensure the continuity of the log harvesting activity, the forest roads must be permanently open and stable. The paper correlated the size of the reconstruction of a forest road with the amount of precipitation, the volume of logs transported (especially high-traffic roads), and its maintenance. Mathisen et al. [11] have studied the influence of forest roads on the habitat of deer in the area, by changing the routes of movement and feeding, the space used, and even mortality. In direct correlation with these, the effect of forest regeneration in the spring season was analyzed. It was observed that in the adjacent area to the forest road the density of undamaged trees was much higher than in the interior areas of the forest. Kazama et al. [12] made an exhaustive study of the main articles published in the period 2009–2019 regarding sustainable forest management in order to optimize the forest road traffic. It is specified the importance of

forest roads, by allowing the harvesting, collection, and sustainable dispatch of some forest natural resources. Aruga et al. [13] developed some equations for estimating the density of the forest road network, before and after mechanization in a mountainous region of Japan. The operational efficiency and the costs for rehabilitating of the forest road network on an area of 6 hectares and a road density of 200 m/ha were also analyzed. Akay et al. [14] dealt with the importance of forest road networks in extinguishing forest wildfires. The emphasis of the research was put on critical response time.

The composition and dimensioning of the component layers of the road structures must be made in such a way as to be able to support the different intensities of traffic, in maximum tonnages or maximum number of trucks. The layers in the composition of a road system are four: the coating, the base layer, the foundation, and the foundation substrate. The clothing or the layer of wear and rolling protects the road structure against the action of atmospheric agents and are made of superior granular materials. The base layer has the role of reducing the concentrated compressive stresses transmitted by the clothing and distributing them to the foundation. The foundation receives the pressures transmitted by the base layer or, in its absence, by the clothing, reduces them even more, and distributes them to the forest bed. Sometimes, road structures do not always cover all layers, nor is there an exact delimitation between them. In the practice of forest road execution, non-rigid road systems are used, adapted to the conditions of forest road traffic. They will have 1–3 road layers, depending on the intensity of traffic, the quality of the stony materials used and the nature of the soil in the forest roadbed (Figure 1).

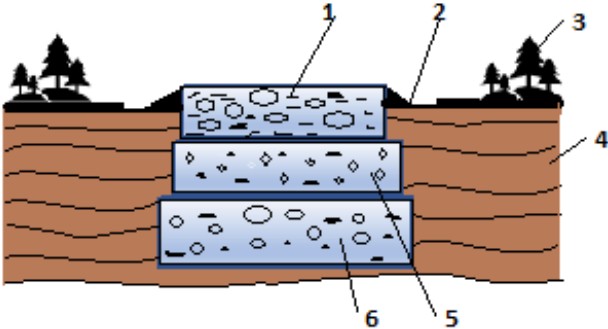

**Figure 1.** Composition of road structures used on forest roads 1—road pavement (wear layer); 2—slope (water collection ditch); 3—forest; 4—forest soil; 5—foundation; 6—substrate foundation.

Objectives. From the bibliographic study, it is observed the tendencies of managerial studies of roads and forest road networks, but their degradation characteristics have not been or very little studied. Therefore, the present study focused on this problem of degradation of a certain forest road and on the methods of qualitative quantification of this property. Tracking the behavior over time of the Ciobanus forest road aims to determine its technical condition between two cycles of maintenance works, identify the causes that led to degradation, and determine in the laboratory or "in situ" the technical, quantitative, and qualitative characteristics of the forest road structure. Consequently, the sizing of maintenance works will be followed, so as to ensure the functionality in safe conditions of the Ciobanus forest road.

## 2. Materials and Methods

All research was carried out on the Ciobanus forest road, the road under the administration of the Bacau Forestry Department, Ciobanus Forest District, Romania. The area where the forest road under study is located was in the Oituz and Trotus Mountains, characterized by a rugged relief, with elongated shells on the north-south direction and with ridges without large unevenness. The valleys that cross these mountains are generally transversal, as is the case of the analyzed valley Ciobanus (Figure 2).

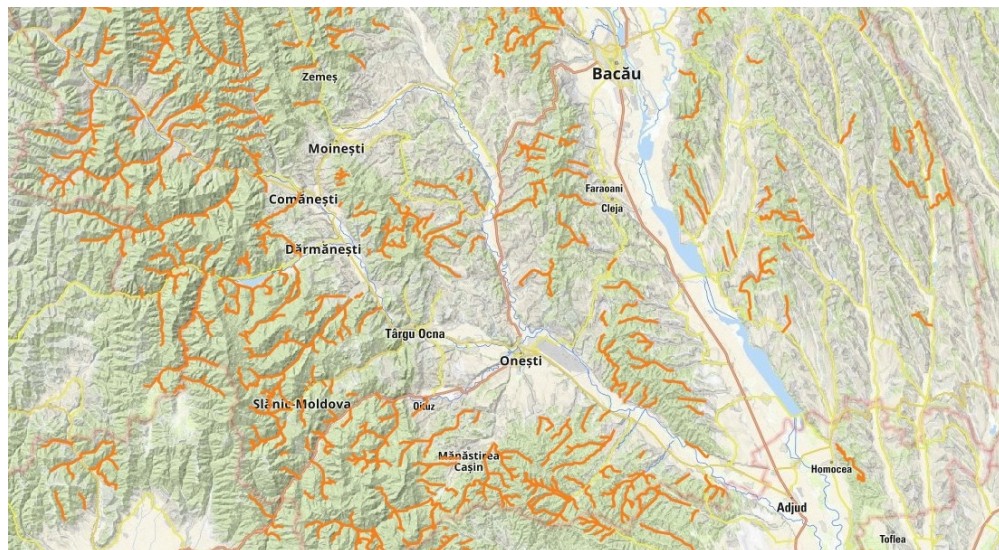

**Figure 2.** Location of the Ciobanus forest road.

Analyzing the relief forms present in the area under study, it is found that the slope shape predominates, mainly medium slopes with 43.7% and higher with 25.5%. Regarding the land configuration, the data collected indicate the predominance of undulating forms, which are found in an overwhelming proportion in the production unit (99%), very few plots having a flat configuration, both forms totaling only 1% of the entire territory. Regarding the land exposure, there is a predominance of lands with southwest exposure (22%) and northeast (17%). If the slope of the land is taken into account, it is found that the trees in the analyzed territory are located on lands with a high slope of over 30% (43%) and very high up to 25% (36%), slightly less being located on lands with a medium slope (16%) and very few on surfaces with a slope below 10 degrees (1%). The altitude is between 460 m and 1375 m, with a mean of 918 m, most of the areas (35%) having altitudes between 701 m and 1100 m. In conclusion, the production unit in which the research was carried out is constituted, in most (99%) of the slopes, as the main relief form, it has landed with predominantly south-western exposure (22%), with a wavy terrain configuration (99%), a predominantly large and medium slope, and altitudes between 460 m and 1375 m, although most areas (76%) are found between 701 m and 1100 m altitude.

15 points on the Ciobanus forest road were taken in the analysis, of which the first point having the coordinates Lat/Lon: 46°26′06.9147″ N, 26°20′29.0629″ E, and the last point of coordinates Lat/Lon: 46°23′00.7369″ N, 26°13′12.8619″ E. The short period between March and June 2018, in which observations and measurements were made, is characterized by high traffic, the long winter period, the appearance of the freeze-thaw phenomenon at the end of the winter season, excess of precipitation during April–June, and also of other phenomena and factors that determine overloads on the structure of this forest road.

Road traffic on the Ciobanus forest road. Road traffic on forest roads was the main cause of their technical wear and tear. Certainly, the technical evolution of the means of transport has been much faster than the development and modernization of the forest road network. In order to highlight the traffic structure, on the Ciobanus forest road, daily traffic monitoring was performed, for a long period of time, more precisely for 5 years (from 2013 up to the first semester of 2017). The daily data collected consisted in mentioning, for each transport of logs, the following information: date, registration number of the platform/trailer, number of approvals accompanying the logs, consignee of transport, assortment transported, mass and volume transported broken down by species (resinous and/or deciduous), mass of the unladen means of transport, and the total tonnage of transport. The collected data were cumulated by months and years, in order to process them statistically and qualitatively.

Dynamic penetration test. All data collection and interpretation procedures complied with the provisions of standard SR EN ISO 22476-2 [15], which indicates the requirements to be taken into account in the case of indirect field tests, through dynamic penetration tests (part of geotechnical tests), using various methods and types of dynamic penetrometers.

Based on SR EN 22475-2, 4 dynamic penetration test procedures are described, depending on the specific impact energy of the blow and, implicitly, on the equipment used. Depending on the mass of the hammer used, there is a light dynamic penetrability with a hammer of 10 kg and a depth of up to 8 m, two intermediate tests with a hammer with an average weight of 25 kg and 50 kg, and a super heavy test with a hammer of 60 kg and a depth of over 25 m. To determine the above elements, 15 dynamic penetration tests were performed, using the Pagani DPM 20–30 dynamic penetrometer (Pagani Geotechnical Equipment, Marsilea, France) with which "in situ" data were obtained.

The dynamic penetration test consisted, in fact, of introducing a conical tip into the field, by making progressive advances and measuring the number of strokes, necessary to insert the cone to a depth of 10 cm. Depending on what the depth of penetration of the penetrometer (cone tip) in the field was, which determined the resistance to dynamic penetration on the cone (Rd), which represents the opposite resistance of the field to advance the penetration cone under the action of constant mechanical work, achieved by the falling ram. The dynamic penetration resistance is determined by the following Equation (1):

$$Rd = 1 : A \times (M^2 \times H) : (e \times (M + P)) \; [\text{kPa}] \tag{1}$$

where: Rd is the dynamic penetration resistance [kPa]; e-the penetration of the cone under a single stroke (average penetration per stroke, resulting as a ratio between the step of the instrument) and the number of strokes; H-drop height of the cone [m]; M-ram weight [kN]; P-total weight of the rod and the striking system [kN]; A-the cross-sectional area of the cone [$m^2$].

The specific energy for each shot is calculated by the Equation (2):

$$Q = (M^2 \times H) : (A \times \delta \times (M + M')) \; [\text{kN}] \tag{2}$$

where: Q is the specific energy for each blow [kN]; M-hammer weight [kN]; M'-rod weight [kN]; H-fall height [m]; A-surface area of the cone cross section [$m^2$] $\delta$-penetration interval (instrument pitch).

The processing of the obtained data was performed with the help of an automatic calculation program-Dynamic Probing (GeoStru Software, Civilserve GmbH, Steinfeld, Germany).

Determination of granularity by sieving. Determination of granularity by sieving is made based on SR EN 933-1:2020 [16]. The principle of the method was to separate the material into several decreasing dimensional categories by passing the sample through a series of sieves. The equipment used consists of a test sieve (square mesh sieve, between 63 mm and 0.063 mm), sieve mounting brackets with lid, oven, washing equipment, scale, and brushes for the sieving machine (optional). The method of sample preparation is regulated by SR EN 933-1 [17] and depends on the maximum diameter of the aggregates.

Determination of granularity by sieving involved a number of activities, such as washing, sieving, and weighing the samples, followed by the calculation and interpretation of the results. The determination of the content of fine particles passed through a sieve with a mesh diameter of 0.063 mm involves the application of Equation (3):

$$f = ((M_1 - M_2) + P) : M_1 \times 100 \; [\%] \tag{3}$$

where: f is the content of fine particles passed through a sieve with a mesh diameter of 0.063 mm [%]; $M_1$-dry mass of the sample [kg]; $M_2$-dry mass retained on the sieve with a mesh diameter of 0.063 mm [kg]; P-mass of residual materials remaining in the vessel.

For dry sieving, Equation (4) was used, where the terms had the same meanings as in Equation (3).

$$f = P:M_1 \times 100 \ [\%] \tag{4}$$

Determination of crushing strength. This test was performed on the basis of SR EN 1091-1 [18], a European standard presenting the methods for determining the crushing strength of coarse aggregates, defining two methods, respectively: the Los Angeles test (reference method) and the impact crushing test (alternative method). In the case of the research undertaken, the Los Angeles test was applied. The necessary equipment consisted of some sieves as EN 933-2 stated [19], with mesh size according to SR EN 1097-2: 2002 [20], balance, oven sample reduction equipment, and Los Angeles test machine (Waterloo, ON, Canada).

The principle of the method was to insert a sample of aggregates into a rotating drum and to roll the aggregates subjected to steel ball tests. In the end, the amount of material retained on the 1.6 mm sieve is determined. Both the sample preparation and working method are described in SR EN 1097-2: 2002 [20]. A temperature-adjustable oven for drying materials (capacity 120 L and ambient temperature range of 5–250 °C) was used.

The Los Angeles coefficient was determined by the relation (5), where *m* represents the mass of the rejection on the 1.6 mm mesh screen, expressed in grams.

$$LA = (5000 - m):50 \tag{5}$$

Determination of the degree of compaction by using the dynamic plate. The simplest method for determining the degree of compaction of embankments was to determine the dynamic deflection modulus using the dynamic plate. The dynamic modulus of linear deformation is the ratio between the unit compression force applied on the test layer and the corresponding specific deformation.

The role of the dynamic plate is to determine, by direct field tests, the linear deformation modulus (dynamic deflection modulus). This method was applied to both non-cohesive and cohesive soils that form part of embankments and foundation or base layers. Thus, the Zorn ZFG 0.2 dynamic board (Zorn Instruments GmbH & Co. KG, Hansestadt Stendal, Germany) is mainly used in the construction of roads or highways, in pavement testing, for testing compaction characteristics in sewers and pillars of foundations. The Zorn ZSG 0.20 dynamic plate that was used, was composed of: the test assembly necessary to produce a defined loading impact (Evd 5–70 MN/m-10 kg), the 300 mm test plate, the ZSG electronic deflectometer 0.2 (Zorn Instruments GmbH & Co. KG, Hansestadt Stendal, Germany), with built-in data memory and digital interface, SD card reader and USB connection, processing software, mini-printer, and batteries.

Compared to the usual method, the determination of the linear deformation model based on STAS 8942/3-90 [21] presupposed a very laborious working method, with complex equipment and numerous data readings, the result not being provided immediately (Equation (6)).

$$EVD = 22.5:S \ [MN/m^2] \tag{6}$$

It should be noted that the other parameters of the plate, namely the plate mass, its diameter, and the height of fall are known and constant, as they are specified in the technical book of the plate. To determine the correspondence between the static strain modulus (EV2) and the dynamic strain modulus (EVD), the Equation (7) was applied:

$$EV_2 = 600 \times \ln f_o \times 300 \times (300 - EVD) \ [MN/m^2] \tag{7}$$

In accordance with the provisions of STAS 8942/3-90, for a compaction degree higher than 99%, the dynamic modulus will be higher than 45 MN/m$^2$, for a compaction degree higher than 98%, the dynamic modulus will be higher as high as 40 MN/m$^2$, and for a compaction degree greater than 97%, the dynamic modulus will be greater than 35 MN/m$^2$.

Index of plasticity. Plasticity is the property of cohesive soils to be shaped, keeping the volume and without losing the continuity of the mass. At a certain moisture variation, the soils go through different physical states (solid, semi-solid, plastic, fluid), being possible to determine the moisture content range at which they behave plastically (SR EN ISO 14688-1). In the case of soils, the property of being plastic can be summarized by five indicators, namely:

- the contraction limit (s), which represents the moisture content below which the decrease of the soil volume does not take place;
- the lower limit of plasticity (p), also called the kneading limit, which indicates the moisture content from which the soil becomes plastic;
- the upper limit of plasticity (f), also called flow limit, which represents the moisture content at which the soil becomes fluid;
- plasticity index (Ip), represented by the interval between the upper limit of plasticity (Wc) and the lower limit of plasticity (Wp), calculated with the Equation (8).

$$I_P = W_c - W_p \ [\%] \tag{8}$$

- the consistency index (Ic), which quantitatively expresses the physical state of cohesive soils and depends on the water content, and can be calculated with the relation (9), where W represents the soil moisture in its natural state.

$$Ic = (Wc - Wp):Ip \tag{9}$$

Soils with a plasticity index of less than 10% have very low plasticity, while those with a plasticity index of more than 50% have high plasticity. The determination of plasticity limits is performed in the laboratory in accordance with the provisions of STAS 1913/4-86, which establishes its own methods of determination.

Determination of lower plasticity limits. The procedure is applied to soils made up of particles less than 2 mm in size and containing organic matter up to 5% by weight, in the dry state, by the soil cylinder method. This method consists in determining the minimum moisture content at which a soil can be shaped into cylinders. From the soil under analysis, a quantity of approximately 100 g is taken, which is sifted, homogenized, and then kneaded. If the sample under analysis does not have sufficient moisture, it can add water until a thick paste is obtained. From the resulting paste, it takes a quantity from which a cylinder with a diameter of 3–4 mm and a length of 30–50 mm is rolled on a sheet of glass. Examine the surface of the earth cylinder, and if no cracks appear on its surface or the cylinder does not break into pieces, the sample is mixed again (to lose water) and rolled until the first cracks appear on the surface of the cylinder or can no longer be made, due to the low humidity that leads to its rupture. This is the moment when, according to STAS 1913/1-82 [22], the moisture in the sample corresponds to the lower limit of plasticity, and the sample can be introduced into the oven to determine the moisture.

Determining the upper limits of plasticity. The procedure applied to soils composed of particles less than 2 mm in size and containing organic matter up to 5% by mass, in the dry state. The cup method and the one-point method consist in determining the moisture content at which a slit made in a paste of soil from the cup of the Casagrande apparatus, closes on a length of 12 mm after 25 falls of the cup from a height of 10 mm. In the case of the cup method, a quantity of 200 g is taken from the soil under analysis and passed through a grater, water is added and is homogenized until a homogeneous paste results. In the case of soils where fractions larger than 0.5 mm predominate, a quantity of approximately 200 g is taken from the soil under analysis and dried in an oven, after which it is ground and passed through a 0.5 mm sieve. Then 2/3 of the Casagrande apparatus (Utest, Ankara, Turkey) cup was filled to determine the plasticity of the soils, its surface is leveled, a deep slit was drawn in the middle of the cup to the bottom of the cup. The bucket is dropped on the base of the device from a fixed height of 10 mm, at constant time intervals (2 drops per second), the device being equipped with a crank connecting mechanism through which the

operator can raise the cup to a height of 10 mm, after which it falls freely on the pedestal. The operation was repeated until the slot closes on a length of 12 mm. The moisture content of the sample is determined, according to STAS 1913/1-82, and the number of cup falls is recorded. The upper limit of plasticity is determined by the next relation (10):

$$W_C = W + K \ [\%] \tag{10}$$

where: $W_C$ represents the upper limit of plasticity [%]; W—moisture content of the sample [%]; K—coefficient determined according to the number of cup falls.

Regarding the statistical processing of the results, the two statistical parameters of trend and spread (median and standard deviation) were used. By using the facilities offered by the Microsoft Excel program, all the charts will have positioned the standard deviation, taking into account a 95% confidence interval of the values. The Minitab 18 statistical analysis program was also used, from which the Empirical Cumulative Distribution Function (CDF) curve was taken in order to establish the normal distribution, the anova-one-way statistical analysis, and the Levene test in order to measure the specific differences between the pair of means.

## 3. Results

Considering the previously mentioned methodology, the technical condition of the Ciobanus forest road can be determined, taking into account the criteria set out in the Norm for the maintenance of forest roads [23].

### 3.1. Traffic on the Ciobanus Forest Road

Traffic on the Ciobanus forest road fluctuated moderately from year to year, but the largest differences were found during each year, with low values at the beginning and end of the year and high values in the middle of the year, as can be seen from Figure 3. Average traffic values were around 30,649 tons of wood/year (or a transited tonnage of 47,180 tons/year, including the means of transport), for the entire period of 2013–2017. The average annual volume of material transported was 34,338 m$^3$/year, which means an average density of logs, for all transported wood species (softwood and hardwood), of about 892 kg/m$^3$ when their real moisture content was taken into consideration.

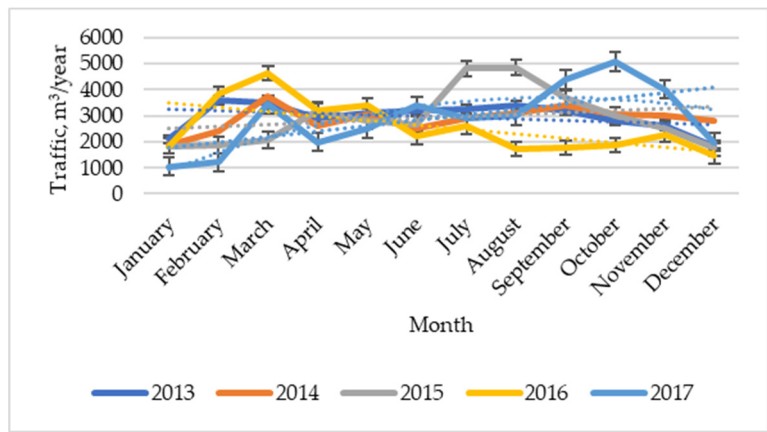

**Figure 3.** Truck traffic on the Ciobanus forest road during 2013–2017.

Taking into account the thickness of the road structure of the road of about 40 cm and in compliance with the traffic regulations (minimal 80 cm depth for maximal 5000 t/year traffic), it can be stated that the traffic on this road is about 10 times higher than the one provided by the regulations. From this point of view, it can be concluded that the Ciobanus forest road must be completely rehabilitated, starting from its infrastructure.

Between March and June 2018, a number of 158 tracks transited on the Ciobănuș forest road, transporting a volume of 5232 m$^3$ of logs, the average volume of a charge

being 33 m$^3$/charge, and the average tonnage of 41 tons/charge. The main degradation observable in this period of time was the erosion of the road surface due to the high humidity of air and moisture content of the soil, an action also observable by the author Aricak [5] in the study conducted on the forest road in the Kastamonu region of Turkey and by Parsokhoo et al. [8].

### 3.2. Granularity of Road Material

The classification of ballast materials used in the case of Ciobanus forest roads is made according to the size of the granules, provenance, processing technology, and granularity, as follows: sand (0–4 mm), gravel (4–31 mm), ballast (0–63 mm) and boulders (63–350 mm). SR EN 13242 + A1 [24] establishes the quality conditions for the ballast used for the maintenance of forest roads. In order to interpret the data, the results of the laboratory analyses will be compared with the permissible values for the ballast categories (ballast, optimal ballast, and ballast for completions), in order to remove the effect of the freeze-thaw phenomenon) provided in the SR 662 regulations [25]. According to the same standard, the size of the predominant fraction (over 50%) is dependent on the soil type, with values of 0.05–0.25 mm for fine sand, 0.25–0.50 mm for medium sand, 0.50–2.0 mm for large sand, 2–20 mm for small gravel, 20–70 mm for large gravel, 70–200 mm for small blocks and over 200 mm for stone blocks. Based on STAS 1243-88 [26], the soils are classified according to the proportion of the majority particle size fraction. The results of the analyzes in the 15 samples classified the soils differently, depending on the size, as follows: 12 in the category "small gravel" 3 in the category "large sand", the predominant particle size fraction being in the range of diameters 2–20 mm, specific to small gravels.

Regarding the granulometric uniformity of the samples, their classification was also based on STAS 1243-88 [26], determining the non-uniformity coefficient, which led to acceptable values of over 15. The graph in Figure 4 shows the granulometry of the material used to make the wear surface of the forest road, by using the 11 sorting sieves, in a cumulative curve. It is clearly observed the framing of the real values between the upper and lower limits. Also, the polynomial regression curves of the cumulative values approximate with high precision the trend of the values, through coefficients of determination R$^2$ of over 0.97 for real granulometry and over 0.99 for lower/upper granulometry. High values of the coefficient demonstrate the uniformity of the values.

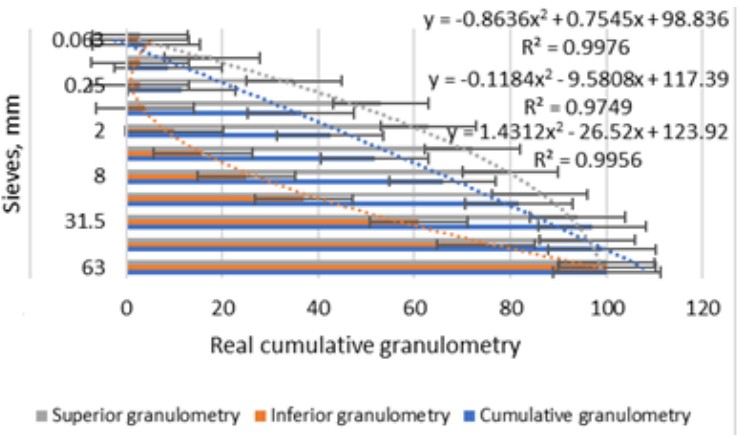

**Figure 4.** Cumulative granulometry of the material used on the Ciobanus forest road.

Taking into account the granulometry of the ballast material that was used, the Ciobanus road corresponds to the norms and the predominant granules fall into the gravel category.

### 3.3. Determination of Compaction Characteristic

Figure 5 shows the correlation diagram between the moisture content and the optimal density for compaction. An optimum moisture content of 4% is observed for the compaction density of 2.224 g/cm$^3$. The regression curve for these values is a 2nd -polynomial grade, with a Pearson determination coefficient of over 0.95, which demonstrates the normal distribution of values for a 95% confidence interval and an alpha error of 0.05 and a homogeneity of values.

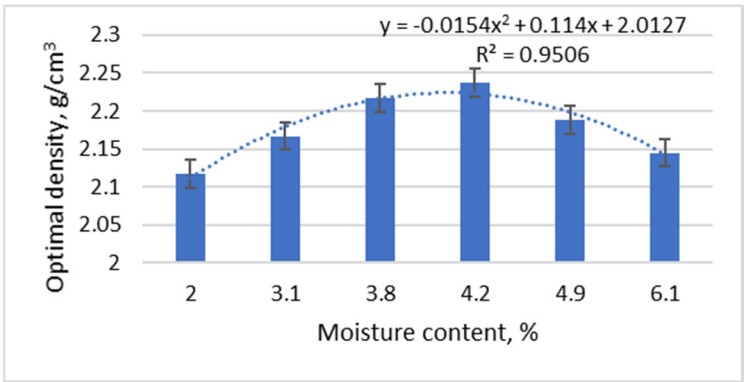

**Figure 5.** Diagram of compaction characteristics for the medium value of material.

The data obtained for the 15 surveys regarding the optimum moisture content and density in the dry state (Figure 6), which can lead to the determination of the effective compaction degree, had values between 97–100%.

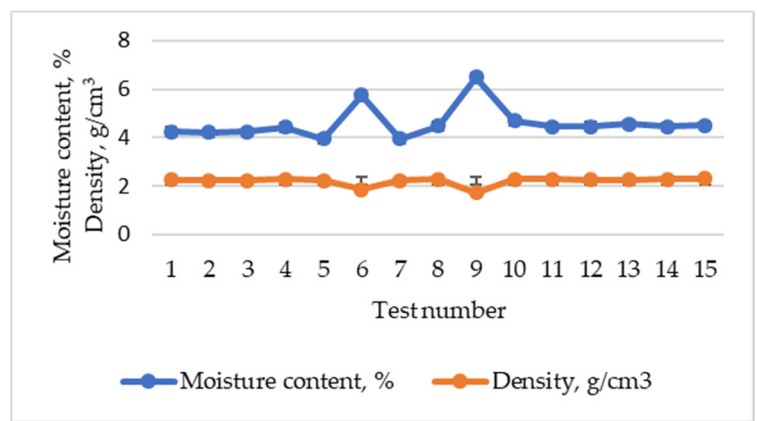

**Figure 6.** Centralization of the data for density and moisture content obtained for the 15 experimental compaction tests.

It is also observed from Figure 6, the points 6 and 9 with maximum values of density of 5.77 g/cm$^3$ and 6.5 g/cm$^3$, corresponding to minimum values of moisture content of 1.862% and 1.731%. The degree of compaction determined in the laboratory and in the field (Zorn SZG 0.20 plate) was similar, without significant differences, and the values ranged between 97–100%, with usual values of 99%.

### 3.4. Dynamic Penetration Test

Following the determinations made with the Pagani DPM20–30 penetrometer in the 15 survey points, the following succession of layers resulted, from surface to depth:

- Layer 1, at 0.4 m, with a density of 2.2 g/cm$^3$ and a dynamic penetration resistance of 45.08 daN/cm$^2$, corresponding to the superstructure of ballast, gravel, small boulders, sands, and clayey clays;

- Layer 2, at 2 m, with a density of 1.86 g/cm$^3$ and a dynamic penetration resistance of 31.36 daN/cm$^2$, corresponding to dusty clay with gravel;
- Layer 3 at 2.5 m, with a density of 2.5 g/cm$^3$ and a dynamic penetration resistance of 64.23 daN/cm$^2$, corresponding to gravel and boulders.

In order to determine the layers of a forest road and their thickness, Pazhouhan et al. [9] used a ground penetration radar and found structures similar to those found in this research. Following the dynamic penetration tests, it resulted that the superstructure of the Ciobanus forest road is of monolayer type, made of ballast, with variable thickness, between 20 and 40 cm. Considering the results of the determinations and corroborating them with the provisions of the Forest Road Design Norm [23], it results that the current structure of the Ciobanus forest road superstructure is similar to that of a secondary forest road, on which a tonnage of maximum 5000 tons is permitted. As the average value of transit on this road is 47,180 t/year, i.e., an increase of 9.4 times, it can be seen that the road does not correspond from this point of view.

### 3.5. Resistance to Crushing and Wear of Bodywork Aggregates

Regarding the crushing capacity of the materials used on the road surface, the Los Angeles coefficient was determined, respectively the percentage of the analyzed sample that passes through the 1.6 mm screen. The obtained values (Figure 7) showed high resistance to crushing being between (20.3–22.6)% with an average of 21.26%, values that fall within the maximum limit of 30% for optimal ballast and materials to complete a forest road, according to the Romanian standard [18,24,25].

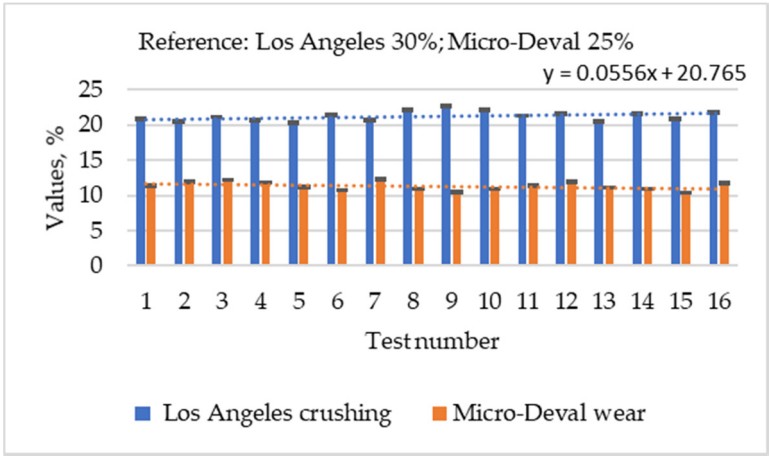

**Figure 7.** Los Angeles and Micro-Deval coefficient of aggregates.

Regarding the wear of the forest road surface, the micro-Deval wear coefficient was determined according to the provisions of SR EN 1091-1 [26], a specific index of ballast and crushed materials currently used on the road of forest roads. This coefficient had values between (10.4–12.3)% and an average value of 11.32%, the admissible limit value being a maximum of 25%.

### 3.6. Particle Size Characteristics

Four dimensional characteristics were determined, respectively the content of fine particles, the sand equivalent, the shape coefficient, and the flattening coefficient, all obtained in the 15 points taken on the route of the Ciobanus road. The average value of the fine particle content was 2.34% in the range of 0.2–5%, with a maximum comparison value of 2%, which means that it is an inappropriate feature of forest mixtures. The average value of the sand equivalent was 52.5%, were between 50.7 and 55.8, compared to the standardized reference of at least 30%, meaning that this characteristic corresponds to the qualitative specifications of the forest roads. The shape coefficient is a specific characteristic

of the river gravel or quarry and of the crushing gravel and has the limit regulated, by SR 662:2002, of at least 25%. The average real value of the shape coefficient was 13.9%, compared to the reference value of at least 25%, which means that this characteristic does not correspond to the qualitative specifications. The flattening coefficient is a distinct property for sieves, crushing sand, and crushed slag material, used for the manufacture of asphalt mixtures, with a regulated limit value of at least 25%. The values obtained in the research did not fall within the limits imposed by the standard, being 15.6% and 20.4%, respectively. The flattening coefficient had an average value of 10.21%, compared to the standardized reference of at least 25%, meaning that it does not correspond to the limiting reference value.

### 3.7. The Degree of Compaction

As shown in Figure 8, which shows the values of the dynamic deformation module for the 15 survey points, there is a large variation in values, which leads to a "sawtooth" curve. The cause of these variations is given by the granulometric non-uniformity of the forest road superstructure, by the moisture content higher than the optimal compaction of this, and, implicitly, by the degree of shading of the area where the data were collected.

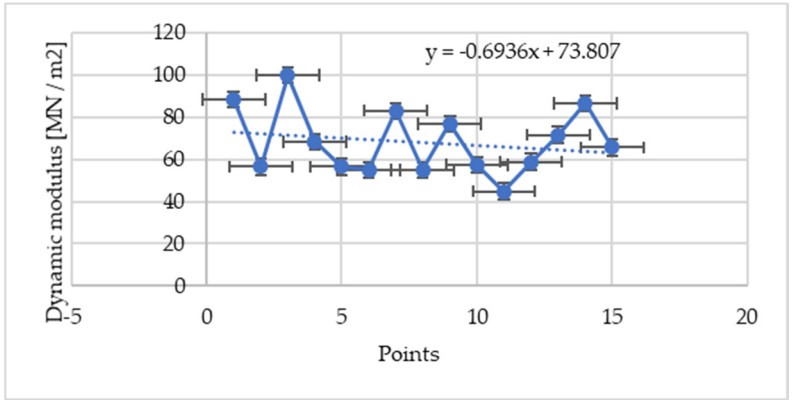

**Figure 8.** Diagram of dynamic deformation modulus.

An average value of dynamic modulus of 68.25 MN/m$^2$ was found (Figure 8), a value much lower than the minimum allowable standard date of 100 MN/m$^2$. This value lower than the minimum reference value makes the forest road not correspond from this point of view. This method of determining the degree of compaction can be very useful in tracking the behavior of forest roads over time, absolutely necessary in the optimal scheduling of maintenance and repair works, but also as a control method at the time of execution of new forest roads, rehabilitation or restoration of objectives affected by disasters. Thus, in the laboratory can be determined the compaction characteristics of the materials to be applied (moisture content, particle size, etc.), and on the field can be checked the quality of the works.

Following the laboratory analyzes, the plasticity characteristics were determined, respectively the lower plasticity limit, the upper plasticity limit (liquidity limit or flow limit), moisture content, non-uniformity coefficient, and, implicitly, the plasticity index. As a result of determining the plasticity index, the plasticity state of the soils could be established, as defined and classified in STAS 1243-88 [27], but also the consistency, according to the same standard. According to this standard, the plasticity index could be zero for no plastic soil, less than 10% for low plasticity soil, 11–20% for medium plasticity soil, 21–35% for soil with high plasticity, and over 35% for soils with very great plasticity. From the point of view of plasticity, a superior average plasticity index of 21.4% was found, which means according to STAS 1243-88 [27], that the wear layer of the Ciobanus forest road has high plasticity in the case of the upper limit and will create landslide problems. The determination of the

plasticity range, as well as the plasticity index, led to the classification of all soils under analysis as having "low plasticity" (Figure 9).

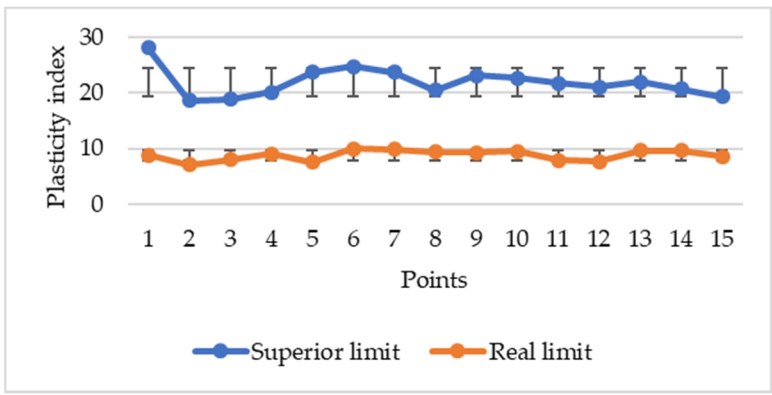

**Figure 9.** Plasticity index for the Ciobanus forest road.

### 3.8. Road Consistency Index

Taking into account that, according to STAS1243-88, the consistency and liquidity index are complementary (their sum is equal to 1), the soils are classified in flowing soil with an index of consistency 0, flowing plastic soil with an index less than 0.25, soft plastic soil with an index of 0.26–0.50, consistent plastic soil with an index of 0.51–0.75, fluffy plastic soil with an index of 0.76–0.99 and hard ground with an index greater than or equal to 1, there were obtained 15 different values shown in Figure 10.

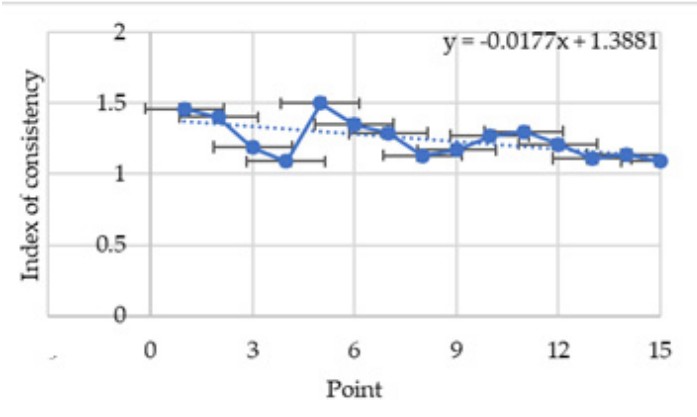

**Figure 10.** Road consistency index.

The mean of consistency index of the aggregates of 1.28, with 28% higher than the recommended value, made it possible to introduce the aggregates used in the "hard ground" category, a material category with a very good consistency (Figure 10), that can withstand the frequent and strong demands to which the road is subjected.

### 3.9. Physical Evaluation of the Forest Road

The physical evaluation of the forest road (unevenness, pits, and degradation), from the point of view of degradations and damages observable or determined visually or with simple measuring means (Table 1), showed the very bad condition of the road, subjected to heavy traffic, much higher than allowed by road category.

### 3.10. General Technical Evaluation of the Forest Road

In view of the general evaluation of the Ciobanus forest road, all the obtained values were taken over and compared with those given by the norms (through maximum and

minimum values or some technical ratings, as is seen in Table 1). Following the visual study of the degradations, all data were taken over and centralized.

**Table 1.** Technical evaluation of the Ciobanus forest road.

| No | Parameters of Evaluation | UM | Technical Status Ratings | | | Real Values | Real Status |
|---|---|---|---|---|---|---|---|
| | | | **Good Satisfactory** | | **Bad** | | |
| 1 | Traffic on crossable | t/year | ≤5000 | - | ≥5000 | 47,180 | I |
| 2 | Granularity | - | Sand | Gravel | Stone | Gravel | S |
| 3 | Dynamic penetration | daN/cm$^2$ | 100 | 90 | 80 | 64.23 | I |
| 4 | Crush resistance | % | ≤30 | - | ≥25 | 21.26 | S |
| 5 | Wear resistance | % | ≤25 | - | ≥25 | 11.32 | S |
| 6 | Fine particle content | % | ≤2 | - | ≥2 | 2.34 | I |
| 7 | Sand equivalent | % | ≥30 | - | ≤30 | 52.5 | S |
| 8 | Shape coefficient | % | ≥25 | - | ≤25 | 15.6 | I |
| 9 | Flat coefficient | % | ≥25 | - | ≤25 | 20.4 | I |
| 10 | Dynamic module | MN/m$^2$ | ≥100 | - | ≤100 | 68.25 | I |
| 11 | Index of plasticity | % | ≤10 | - | ≥10 | 21.4 | I |
| 12 | Consistency index | - | ≥1 | - | ≤1 | 1.24 | S |
| 13 | Bearing coefficient | % | ≥85 | 70–85 | ≤70 | 90 | I |
| 14 | Wear coefficient | % | ≤20 | 20–35 | ≥35 | 48% | I |
| 15 | Med. elastic deform. | mm | 2.2–2.5 | 2.5–3.0 | ≥3.0 | 10 | I |
| 16 | Degradations | pieces/km | ≤100 | 100–160 | ≥160 | 227 | I |
| 17 | Damaged depth, medium | cm | 2 | 3 | 4 | 10 | I |
| 18 | Damaged depth, maximum | cm | 4 | 6 | 8 | 30 | I |
| 19 | Damaged area | m$^2$/km | ≤75 | 75–225 | ≥225 | 1483 | I |
| 20 | Damaged area | % | ≤2.5 | 2.5–7.5 | ≥7.5 | 49 | I |

S—Suitable, I—Improperly.

It is observed that 80% of the criteria are inappropriate, which means that the forest road does not correspond to the minimum of 80% elements from all 20 analyzed ones. Applying the criteria for assessing the technical condition of the roads resulted in the Ciobanus forest road having inadequate technical conditions, and which, as is the norm for the maintenance of forest roads [28] recommended in this case, needing the complete rehabilitation of the forest road. This rehabilitation must be in accordance with the traffic on this forest road with capacities of over 30,000 t/year, thus transforming a secondary forest road into a principal one.

## 4. Discussion

The sharp increase in the volume transported (34%), coupled with the less aggressive increase in tonnage per journey (20%), is an indication of the renewal of the fleet of trucks used in log transport, but also increased efficiency in managing the costs of logs mass transport. As above, the authors Cavalli and Grigolato [6] and Akay et al. [10] show that the more intense the degradation of the forest road, the higher the transport costs will be.

Table 2 presents a One-way ANOVA statistical analysis of wear resistance, expressed by Los Angeles and Micro Deval coefficients. The null hypothesis is taken into consideration, namely, all means are equal. It can be observed that the *p*-value of $6.00014 \times 10^{-12}$, is much less than the alpha error value of 0.05. From this above observation, it can be concluded that the distribution of values is normal, meaning that the null hypothesis is rejected.

**Table 2.** ANOVA statistical analysis for wear resistance.

| Source of Variation | SS | df | MS | F | *p*-Value | F Crit |
|---|---|---|---|---|---|---|
| Between Groups | 16,085.54 | 1 | 16,085.54 | 127.81 | $6.00014 \times 10^{-12}$ | 4.1959 |
| Within Groups | 3529.876 | 28 | 125.85 | | | |
| Total | 19,608.43 | 29 | | | | |

In correlation with the results, De Witt et al. [4] found that the wear resistance of the road surface depends on the nature of the granular materials, making a prediction of this mechanical property depending on the materials used.

By using the facilities offered by the statistical program Minitab 18, an Empirical CDF (Cumulative Distribution Function) curve was made (Figure 11), from which it can be to observe, besides the approximation of the real points by their normal empirical distribution, small values of the standard deviation of 0.67 and 0.59 for Los Angeles and Micro Deval coefficients, respectively.

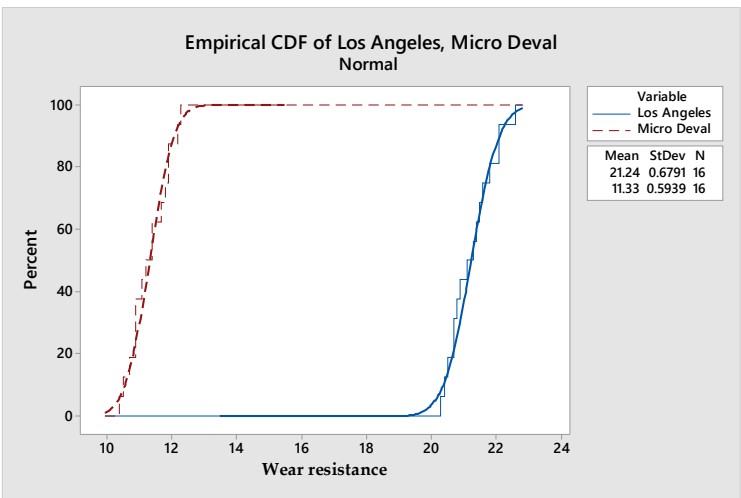

**Figure 11.** Empirical CDF of wear resistance.

Assuming alternative hypothesis, namely not all means were equal or when was a difference in means, the test for equal variance, namely Levene test (with multiple comparisons) was used in the case of wear resistance (by Los Angeles and Micro Deval coefficients), obtained a *p*-Value of 0.0428. This value was compared with the alpha error of 0.05 and found that all values are in the confidence interval of 95% for normal distribution, respectively in the limits of ±2 s, where s is the standard deviation of results.

The analysis of the degradation of the Ciobanus forest road had practical importance by establishing the degree of degradation and the immediate measures that are challenged. The efficiency of the rehabilitation of a forest road was also shown by other authors as Aruga [13], determining a higher density of trees around forest road [11], with a decisive impact on hydrological processes [1], on wildfire management [2], harvesting the feller or sick trees [3], and ecological management [7].

## 5. Conclusions

The number of tracks that transited the Ciobanus forest road experienced a slow but continuous decrease, but the decrease of the volume and the transported mass was much smaller, due to the increase of the transport capacity of the used trucks. Consequently, increasing the average tonnage, led to a sharp degradation of the road, this being observable from the tests performed, especially those of dynamic penetration.

Corroborating the monthly distribution of logs transport on the Ciobanus forest road, during March–June 2018 with the data of the main climatic elements registered during

the same year, it can be stated that the transport with the highest intensity takes place in October–March, characterized, from a climatic point of view, by low temperatures, relatively high humidity, and days with high wind speeds. Also, towards the end of this interval, the forest road is affected by the freeze-thaw phenomenon. All these elements have led to the accentuated degradation of the forest road and the need for its urgent rehabilitation.

The research has made an exhaustive analysis of Ciobanus forest road, taking into account a series of laboratory or field tests that have been compared with the limit values of international and national legislation, in order to establish the diagnostic and rehabilitation measures.

As results from the laboratory or in situ determinations show, and as performed on the material taken from the structure of the Ciobanus forest road and from the visual follow-up of the behavior in time, it is necessary to completely rehabilitate its structure.

**Author Contributions:** Conceptualization, A.L. and I.B.; methodology, I.B.; software, R.D.; validation, V.C., C.S. and A.L.; formal analysis, R.D.; investigation, I.B.; resources, I.B.; data curation, C.S.; writing—original draft preparation, A.L.; writing—review and editing, A.L.; visualization, R.D.; supervision, V.C.; project administration, A.L.; funding acquisition, C.S. All authors have read and agreed to the published version of the manuscript.

**Funding:** This research received no external funding.

**Institutional Review Board Statement:** Not applicable.

**Acknowledgments:** We would like to thank the Transilvania University of Brasov for all the support provided in conducting the research and drafting the paper.

**Conflicts of Interest:** The authors declare no conflict of interest.

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
