# Peer review of "Research on Tracking the Behavior of the Ciobanus Forest Road over a Season Time through Specific Tests and Analysis"

_applsci, doi:10.3390/app12010459_

Round 1

Reviewer 1 Report

The paper can be accepted now.

Author Response

English has been substantially improved throughout the revised paper.

Reviewer 2 Report

The last paragraph on P.2 is overly long and should be split up. A great deal of content is presented requiring a little more contextual commentary. This paragraph in my view could could be broken up into 3 paragraphs. This will give greater contextual clarity. 

Commentary starting at the top of P.9 makes reference to "collaborating' with traffic regulations. This is the wrong word. It really should be something like in 'compliance' of or something like that. You do not collaborate with regulations you comply with them or not as the case may be. I also think, despite this not being a legal paper, some explanations of the regulations and what they require is probably warranted in order to explain why the authors comment on the road not being in compliance.

The conclusion is my view is not adequate and could be expanded on. The very last sentence in particular refers to ' a general conclusion' but then fails to actually give one. I do believe this can usefully be expanded on.

Author Response

We would like to thank the reviewer for his work in improving the work, in order to increase the visibility for the readers.

-That very long paragraph was divided into 3 parts

-In the paragraph on page 9 “collaborating”, the word in question was changed to "compliance".

-The conclusions have been changed. Some general conclusions were definitively erased, others of these that were inconsistent were supplemented with the data of a good agreement.

See also the attach

Authors

This manuscript is a resubmission of an earlier submission. The following is a list of the peer review reports and author responses from that submission.

Round 1

Reviewer 1 Report

The authors have studied the behavior of the Ciobanus forest road through specific tests over a certain period of time in order to diagnose the degree of wear and degradation. The results showed that 5 of them were classified as appropriate and 15 unsuitable for traffic. The data used in the paper is convincing and the method is adequately described. Furthermore, the conclusions are reliable.
In general, the paper is well-organized and is acceptable after a minor revision.
1. The introduction should be adequate to indicate the general background and to summarise previous work. Specially, the problems to current research should be addressed.
2. Some format problems should be revised, such as Table 1.

Author Response

Reviewer 1

We would like to thank the reviewer of our paper for his work and the pertinent observations and recommendations made.

  1. Reviewer requirement: The introduction should be adequate to indicate the general background and to summarize previous work. Specially, the problems to current research should be addressed.

Author answer: The introductory part has been simplified, giving up a lot of paragraphs. Also, a synthesis of the bibliography was made (in the “objectives” zone), in order to highlight the gaps in the referenced studied field and to explain the need to carry out the paper study.

  1. Reviewer requirement: Some format problems should be revised, such as Table 1.

Author answer:  Table 1 has been reformatted, in accordance with the requirements of the journal and reviewer.

Authors

Reviewer 2 Report

I would like to congratulate the authors for this work. My suggestion to the authors is to make the introduction a little more concise and add a google earth image that shows the roads where the work took place. Also, I think it would be more appropriate to use "pass" instead of "race".

Author Response

Reviewer 2

We would like to thank the reviewer of our paper for his work and the pertinent observations and recommendations made.

  • Reviewer requirement: I would like to congratulate the authors for this work. My suggestion to the authors is to make the introduction a little more concise and add a google earth image that shows the roads where the work took place. Also, I think it would be more appropriate to use "pass" instead of "race".
  • Author answer:
  1. The introductory part has been simplified by deleting many paragraphs. Also, a synthesis of the studied bibliography was made, which would highlight the research opportunity in this paper.
  2. The imagine of Ciobanus forest road was introduced as Figure 2.
  3. The word “pass” was inserted instead of the word “race” where necessary. In some places the word "race" has been replaced by the word "charge", depending on the meaning of the phrase.

Authors 

Reviewer 3 Report

The authors submitted the manuscript to the journal in which they describe "Research on tracking the behavior of the Ciobanus forest road over a season time through specific tests and analysis". The entire article is about 22 pages of content, but nothing innovative is written in it. The manuscript is a to long. 

The discussion was not conducted in accordance with the guidelines of journal.

There is no in-depth statistical analysis. Statistical analysis was not sufficiently discussed.

An expansion of literature, conclusions and discussions is required.

In this form, I advise against publishing the article in this journal.

English needs a lot of improvement.

Author Response

Reviewer 3

We would like to thank the reviewer of our paper for his work and the pertinent observations and recommendations made.

  1. Reviewer requirement: The authors submitted the manuscript to the journal in which they describe "Research on tracking the behavior of the Ciobanus forest road over a season time through specific tests and analysis". The entire article is about 22 pages of content, but nothing innovative is written in it. The manuscript is a to long.

- The whole paper has been simplified, eliminating a series of theoretical paragraphs from all parts of the paper, respectively from the introduction, methodology, results and conclusions. - In presenting the methodology and results in the new version of the paper, we tried to emphasize the original part of the research; - The paper was significantly reduced to 19 and a half pages.

  1. The discussion was not conducted in accordance with the guidelines of journal.

Author answer:  The discussion part has been moved to a separate chapter, "4. Discussion"

  1. There is no in-depth statistical analysis. Statistical analysis was not sufficiently discussed.

Author answer: A paragraph regarding the statistical processing of the obtained values was introduced in the chapter of methodology and materials. All the values used in the paper represented the median of the experimental values obtained in the research. Also, all the diagrams made with the Microsoft Excel program were completed with the standard deviation, as the main statistical parameter of the group of analysed values. 

  1. An expansion of literature, conclusions and discussions is required.

Author answer: The discussions were set out in a separate chapter of the paper, although they were a little bit consistent. The paper was based on the comparison of existing standards in the field, because articles with the exclusive theme of paper did not exist. A new analysis of the studied field was made in the Springer and Elsevier databases, but all the bibliographic titles were found adjacent to the analysed topic, reason for which we did not make changes in the studied bibliography. 

  1. In this form, I advise against publishing the article in this journal.

Author answer: Our work has been substantially improved over the previous manuscript (the theoretical passages were removed and new ones with practical applicability were introduced), which is why, with respect of your last consideration, please reconsider your decision. 

  1. English needs a lot of improvement.

Author answer: All English words and expressions were improved by a native English person.

Authors

Round 2

Reviewer 3 Report

To old literature!

The manuscript has been significantly modernized, but the substantive part still does not convey any significant message. The manuscript does not have any advanced statistical analysis and correlation e.g. with the use of ANOVA with the Duncan test. It will certainly enrich the manuscript.

Author Response

We would like to thank the reviewer for the pertinent observations made on the paper. In the following, the authors will make clarifications regarding the changes brought to the paper, which are in accordance with the reviewer's observations.

Reviewer observations: The manuscript has been significantly modernized, but the substantive part still does not convey any significant message. The manuscript does not have any advanced statistical analysis and correlation e.g. with the use of ANOVA with the Duncan test. It will certainly enrich the manuscript.

Author answers:

  • Lines 412-415: In the area of statistical processing of results, a new paragraph has been introduced, which explains the use of the statistical processing program Minitab 18 together with the facilities offered by it. Among these facilities, the Analysis of Variants (ANOVA), the CDF graph and the Levene test were taken. Among the tests that measure the specific differences between pairs of means such as Tukey, t-student, Duncan Multiple Range, Newman-Keuts, Fisher LSD, etc., the Levene test was chosen because it belongs to the Minitab 18 program, with which the authors are familiar.
  • Lines 634-638: In this area is explained the newly introduced table number 10, referring to ANOVA statistical analysis, regarding the normality of the distribution of values and the P-Value value.
  • Line 640: Table 10, with ANOVA analysis was introduced.
  • Lines 645-648: In this area, the specifications regarding the newly introduced CDF chart are made, for statistical data processing and verification if the distribution of values is normal.
  • Line 651: In this area the graph of figure 12 was introduced.
  • Lines 653-658: in this area the explanation of Levene test was introduced
  • Line 684-685: In the zone of conclusion, a new general conclusion was introduced. This new conclusion tries to make sense and substance of the whole research.
  • To all this is added what was introduced in the first revision, respectively the introduction in each diagram of the standard deviations for each point of the diagrams, as well as the respective methodological specifications.